# Supersymmetric ground states of 3d $\mathcal{N} = 4$ SUSY gauge theories and Heisenberg algebras

**Andrea E. V. Ferrari**$^\star$

Department of Mathematical Sciences, Durham University,
Upper Mountjoy Campus, Stockton Rd, Durham DH1 3LE

$\star$ andrea.e.v.ferrari@gmail.com

## Abstract

We consider 3d $\mathcal{N} = 4$ theories on the geometry $\Sigma \times \mathbb{R}$, where $\Sigma$ is a closed and connected Riemann surface, from the point of view of a quantum mechanics on $\mathbb{R}$. Focussing on the elementary mirror pair in the presence of real deformation parameters, namely SQED with one hypermultiplet (SQED[1]) and the free hypermulitplet, we study the algebras of local operators in the respective quantum mechanics as well as their action on the vector space of supersymmetric ground states. We demonstrate that the algebras can be described in terms of Heisenberg algebras, and that they act in a way reminiscent of Segal-Bargmann (B-twist of the free hypermultiplet) and Nakajima (A-twist of SQED[1]) operators.

 Check for updates

# 1  Introduction

3d $\mathcal{N} = 4$ gauge theories and their infrared dualities, known as 3d mirror symmetry, have turned out to be a rich playground for researchers interested in geometry and representation theory. For example, the study of the vacuum structure of mirror dual theories has inspired a conjectural mathematical duality known as symplectic duality, which in its simplest form relates deformations and hamiltonian isometries of two different hyperkähler manifolds. This relation can be generalized by studying the mathematical definitions and properties of more sophisticated physical observables [1–5].

In the last few years, some attention has been paid to 3d $\mathcal{N} = 4$ theories topologically twisted on $\Sigma \times \mathbb{R}$ or $\Sigma \times \mathbb{R}^+$ where $\Sigma$ is a closed and connected Riemann surface, not least because of its relation to the Geometric Langlands Program [6–9]. One basic observable in this set-up is the space of supersymmetric ground states of the 3d theory on $\Sigma \times \mathbb{R}$, which can be studied from the point of view of a supersymmetric quantum mechanics on $\mathbb{R}$ [10, 11] and refines twisted indices [12–17]. Another related observable comes from the study of deformed $(0, 4)$ boundary conditions on $\Sigma \times \{0\} \subset \Sigma \times \mathbb{R}^+$ that support non-unitary Vertex Operator Algebras [18–20].

The aim of this note is to highlight a simple representation-theoretic structure underpinning the vector spaces of supersymmetric ground states in the presence of generic real deformation parameters, which are introduced to ensure the gapness of the underlying supersymmetric quantum mechanical systems. The algebra in question is the algebra of local operators in the quantum mechanics, which contains local operators in 3d as well as their descendants along $\Sigma$. Focussing on the simplest mirror pair, namely a free hypermultiplet and SQED[1], we show that on very general grounds these algebras organise themselves into Heisenberg algebras that are governed by the intersection pairing on $\Sigma$ as well as the secondary product defined in [21]. The spaces of supersymmetric ground states can be understood as Fock spaces for these Heisenberg algebras.

Mirror symmetry relates two different topological twists, known as A- and B-twist, which give the mirror, isomorphic Fock spaces distinct and interesting incarnations. The Hilbert space of a free hypermultiplet in the B-twist is reminiscent of a Segal-Bargmann space [22, 23] associated with the space of constant maps from the curve $\Sigma$ to the Higgs branch $M_H = T^\vee \mathbb{C}$. In fact, the Fock space is generated by creation and annihilation operators that act by multiplication and differentiation on certain bosonic and fermionic polynomial functions that are

square-integrable with respect to a Gaussian measure.

The Hilbert space in the A-twist of SQED[1] can be understood in terms of the de Rham cohomology of vortex moduli spaces, on which monopole operators and their descendants act by creating and annihilating vortices along prescribed cycles on $\Sigma$. This action is the analogue for the symmetric product of a curve of the action introduced by Nakajima on the Hilbert scheme of points on surfaces [24], and it is reminiscent of the Fock spaces associated to symmetric products that appeared in the physics literature in [25]. The B-twist of SQED[1] and the A-twist of the free hypermultiplet become interesting only upon turning on background superfields, something that we only briefly mention here. This combination has already been studied from a mathematical point of view in [4].

Besides being interesting in their own right, we hope that these actions (when properly extended to more general theories, and to the presence of line operators) will help us to understand the relation between boundary conditions, supersymmetric ground states, and conformal blocks. We leave this to future work.

This paper is organised as follows. We first review the topological twists and the effective supersymmetric quantum mechanics on $\mathbb{R}$, and derive the Heisenberg algebras from the commutators of local operators and their descendants. We then describe the action of these algebras in the B-twist of the free hypermultiplets and in the A-twist of SQED[1] in turns.

## 2 SUSY Quantum Mechanics from 3d $\mathcal{N} = 4$ theories

We start by introducing the topological twists, emphasizing some algebraic facts. We work in euclidean signature. The supersymmetry algebra of a 3d $\mathcal{N} = 4$ gauge theory in flat space reads

$$\left\{ Q_\alpha^{A\dot{A}}, Q_\beta^{B\dot{B}} \right\} = \epsilon^{AB}\epsilon^{\dot{A}\dot{B}}P_{\alpha\beta} - \epsilon^{AB}\epsilon_{\alpha\beta}Z^{\dot{A}\dot{B}} - \epsilon^{\dot{A}\dot{B}}\epsilon_{\alpha\beta}Z^{AB}, \tag{1}$$

where $P_{\alpha\beta}$ are the momenta and $Z^{AB}$, $Z^{\dot{A}\dot{B}}$ are central charges.[1] Throughout this paper, the only non-zero central charges will be $Z^{12} = Z^{21}$, $Z^{\dot{1}\dot{2}} = Z^{\dot{2}\dot{1}}$.

The central charges break the R-symmetry to a maximal torus $U(1)_H \times U(1)_C \subset SU(2)_H \times SU(2)_C$. The unbroken R-symmetry is still sufficient to perform a topological twist on $\Sigma \times \mathbb{R}$, where $\Sigma$ is a compact, connected and closed Riemann surface. The first step corresponds to identifying $\mathbb{R}^3 \cong \mathbb{R}^2 \times \mathbb{R}$, and to singling out the $U(1)$ subgroup of the isometry group of $\mathbb{R}^3$ that rotates the $\mathbb{R}^2$ factor. We call this subgroup $U(1)_L$. The second step corresponds to selecting a $U(1) \subset U(1)_H \times U(1)_C$ subgroup and to mix it with $U(1)_L$ to defined an "improved" isometry group. We consider two twists that mix $U(1)_L$ with a subgroup $U(1)_H \subset SU(2)_H$ and $U(1)_C \subset SU(2)_C$ respectively, the Rozanksy-Witten twist and its mirror.

### 2.1 A-twist

The mirror Rozansky-Witten twist will be dubbed "A-twist". Let

$$U(1)_A \subset U(1)_L \times U(1)_H \tag{2}$$

be the diagonal subgroup. The nilpotent combination of supercharges

$$Q_{mRW} := Q_1^{1\dot{1}} + Q_2^{2\dot{1}} \tag{3}$$

is a scalar with respect to $U(1)_A$. Furthermore, if we set

$$Q_{\alpha\beta} := -\epsilon_{(\alpha\gamma}Q_{\beta)}^{\gamma\dot{2}}, \tag{4}$$

---

[1]Our convention for the Levi-Civita tensors is $\epsilon^{12} = \epsilon_{21} = 1$.

where $(\alpha \cdots \beta)$ denotes symmetrisation in $\alpha$ and $\beta$, then

$$\{Q_{mRW}, Q_{\alpha\beta}\} = P_{\alpha\beta} + Z^{\alpha\beta}, \tag{5}$$

with $P_{12}$ corresponding to translations along the $\mathbb{R}$ direction, and we recall that $Z^{\dot{A}\dot{B}} \neq 0$ only for $\dot{A} \neq \dot{B}$. Thus, if we consider a modification of the Lorentz group where $U(1)_L$ is replaced by (2) and work in the cohomology of the scalar, nilpotent supercharge $Q_{mRW}$, translations along $\mathbb{R}^2$ become exact. In particular, in $Q_{mRW}$-cohomology we can define the theory on $\Sigma \times \mathbb{R}$, for $\Sigma$ a compact, connected and closed Riemann surface.

The constituents of the mirror Rozansky-Witten supercharge (3) are a subset of four supercharges that are scalars under $U(1)_A$

$$Q^{\dot{A}} := Q_1^{1\dot{A}}, \quad \widetilde{Q}^{\dot{A}} := Q_2^{2\dot{A}}. \tag{6}$$

They satisfy the algebra of a $\mathcal{N} = 4$ supersymmetric quantum mechanics

$$\{Q^{\dot{A}}, \widetilde{Q}^{\dot{B}}\} = \epsilon^{\dot{A}\dot{B}}\left(P_{12} + Z^{12}\right) + Z^{\dot{A}\dot{B}}, \tag{7}$$

with all other anti-commutator vanishing. Their conjugates are $(Q^{\dot{1}})^\dagger = \widetilde{Q}^{\dot{2}}$ and $(Q^{\dot{2}})^\dagger = -\widetilde{Q}^{\dot{1}}$. In this paper, we shall be interested in the space of supersymmetric ground states of this quantum mechanics, i.e. the states annihilated by the four supercharges $Q^{\dot{A}}, \widetilde{Q}^{\dot{B}}$. It is easy to see that such states must be annihilated by $Z^{12}$. Then, noting that

$$Q_{mRW}^\dagger = Q_1^{1\dot{2}} - Q_2^{2\dot{1}}, \tag{8}$$

we compute

$$\{Q_{mRW}, Q_{mRW}^\dagger\} = 2\left(P_{12} + Z^{12}\right). \tag{9}$$

Thus, provided the spectrum of the operator on the right hand side is gapped, by a standard argument supersymmetric ground states can be identified with states that vanish under the action of $Z^{\dot{A}\dot{B}}$ and that are in the cohomology of $Q_{mRW}$, which means that are well-defined states in the twisted theory.

Since we have an unbroken $U(1)_H \times U(1)_C$ at our disposal, states possess a $\mathbb{Z} \times \mathbb{Z}$ grading. Following [11], we define a new $\mathbb{Z} \times \frac{1}{2}\mathbb{Z}$ grading on the Hilbert space of states by declaring that if $r_H, r_C$ are the eigenvalues of a generator of $U(1)_H, U(1)_C$ then

$$f := r_C, \tag{10}$$

$$r := \frac{1}{2}(r_C - r_H). \tag{11}$$

The action of $Q_{mRW}$ respects the first grading (it sends $f \mapsto f + 1$), whereas it breaks the second. For this reason, we call the first the "primary", or "cohomological" grading and we denote it by $F$. The second grading will be called "secondary" and it will be denoted by $R$. The graded vector space generated by a state with $F, R$ grading $f$ and $r$ will in turn be denoted as follows

$$t^r \mathbb{C}[-f]. \tag{12}$$

## 2.2 B-twist

The Rozansky-Witten twist will be dubbed "B-twist". Let

$$U(1)_B \subset U(1)_L \times U(1)_C \tag{13}$$

be the diagonal subgroup. The nilpotent combination of supercharges

$$Q_{RW} := Q_1^{1\dot{1}} + Q_2^{1\dot{2}} \tag{14}$$

is a scalar with respect to $U(1)_B$. Furthermore, if we set

$$Q_{\alpha\beta} := -\epsilon_{(\alpha\gamma}Q_{\beta)}^{2\gamma}, \tag{15}$$

then

$$\{Q_{RW}, Q_{\alpha\beta}\} = P_{\alpha\beta} + Z^{\dot{\alpha}\dot{\beta}}, \tag{16}$$

Thus, if we replace $U(1)_L$ by (13) and work in the cohomology of the scalar, nilpotent supercharge $Q_{RW}$, we can define the theory on $\Sigma \times \mathbb{R}$.

The constituents of the Rozansky-Witten supercharge (14) are a subset of four supercharges that are scalars under $U(1)_B$

$$Q^A := Q_1^{A\dot{1}}, \quad \widetilde{Q}^A := Q_2^{A\dot{2}}. \tag{17}$$

They satisfy the algebra of a $\mathcal{N} = 4$ supersymmetric quantum mechanics

$$\{Q^A, \widetilde{Q}^B\} = \epsilon^{AB}\left(P_{12} + Z^{\dot{1}\dot{2}}\right) + Z^{AB}, \tag{18}$$

with all other anti-commutator vanishing. The conjugate supercharges are $(Q^1)^\dagger = \widetilde{Q}^2$ and $(Q^2)^\dagger = -Q^{\widetilde{1}}$. As in the A-twist, under some obvious assumptions on the gap of $\{Q_{RW}, Q_{RW}^\dagger\}$, the supersymmetric ground states can be identified with states that vanish under the action of $Z^{AB}$ and that are in the cohomology of $Q_{RW}$.

Finally, we define once again a "primary" or "cohomological" $\mathbb{Z}$-grading $F$ and a "secondary" grading $\frac{1}{2}\mathbb{Z}$-grading $R$ on the Hilbert space as follows. If $r_H$, $r_C$ are the eigenvalues of a generator of $U(1)_H$, $U(1)_C$ then

$$f := r_H, \tag{19}$$

$$r := \frac{1}{2}(r_H - r_C). \tag{20}$$

The action of $Q_{RW}$ preserves the first grading (it sends $f \mapsto f + 1$), whereas it breaks the second. The graded vector space generated by a state with $F$, $R$ grading $f$ and $r$ will in turn be denoted as follows

$$t^r \mathbb{C}[-f]. \tag{21}$$

## 2.3 Descendants and secondary product

Local operators in a cohomological TQFT such as 3d theories A- or B-twisted on $\Sigma \times \mathbb{R}$, which will be our focus, form an algebra induced by the associative product

$$[[\mathcal{O}_1(x_1)]] \star [[\mathcal{O}_1(x_2)]] := [[\mathcal{O}_1(x_1)\mathcal{O}_1(x_2)]]. \tag{22}$$

Here we use $[[\mathcal{O}(x)]]$ to denote the class of an operator $\mathcal{O}$ inserted at a point $x$ in the cohomology of an odd nilpotent operator[2] $Q$. We will drop this piece of notation in later sections, where it will be clear that we will be working in cohomology.

In [21] it was shown that this algebra can be endowed with a Poisson bracket, the so-called secondary product, by means of the topological descent procedure described in [26] that works as follows. Let

$$Q_\mu := -\frac{i}{2}\left(\sigma_\mu\right)^{\alpha\beta}Q_{\alpha\beta}, \tag{23}$$

---

[2]We assume that the configuration space of two points in the manifold where the TQFT is defined is connected, wich justifies the use of the definite article "the".

with $Q_{\alpha\beta}$ the supercharge defined in (4) in the A-twist and in (15) in the B-twist, and $(\sigma_\mu)_\alpha^\beta$ the standard Pauli matrices

$$\sigma_1 = \begin{pmatrix} 0 & 1 \\ 1 & 0 \end{pmatrix}, \quad \sigma_2 = \begin{pmatrix} 0 & -i \\ i & 0 \end{pmatrix}, \quad \sigma_3 = \begin{pmatrix} 1 & 0 \\ 0 & -1 \end{pmatrix}. \tag{24}$$

For $\mathcal{O}(x)$ a local operator and $\mathcal{C}$ a $k$-cycle in the homology of the space-time manifold we define

$$\mathcal{O}^{(k)} := \frac{1}{k!} Q_{\mu_1} \cdots Q_{\mu_k} (\mathcal{O}(x)), \tag{25}$$

$$\mathcal{O}(\mathcal{C}) := \int_\mathcal{C} \mathcal{O}^{(k)} dx^{\mu_1} \wedge \cdots \wedge dx^{\mu_k}. \tag{26}$$

In the absence of central charges the operator $\mathcal{O}(\mathcal{C})$ is topological, because acting upon it with the cohomological supercharge (the one whose cohomology defines the twisted theory) gives the integration of a total derivative over a cycle. In the presence of central charges $Z^{12}$, $Z^{21}$, $Z^{\dot{1}\dot{2}}$, $Z^{\dot{2}\dot{1}}$ on $\Sigma \times \mathbb{R}$, operators of the form $\mathcal{O}(\mathcal{C}_\Sigma \times \{y\})$ for

$$\mathcal{C} \cong \mathcal{C}_\Sigma \times \{y\} \in H_\bullet(\Sigma, \mathbb{Z}) \times H_0(\mathbb{R}, \mathbb{Z}) \subset H_\bullet(\Sigma \times \mathbb{R}, \mathbb{Z}), \tag{27}$$

remain topological.

We can now define the secondary product. Given two local operators $\mathcal{O}_1(x_1)$ and $\mathcal{O}_2(x_2)$, we define

$$(\mathcal{O}_1 \boxtimes \mathcal{O}_2)^{(k)}(x_1, x_2) := \sum_{n=0}^k (-1)^{(k-n)f_1} \mathcal{O}_1^{(n)}(x_1) \wedge \mathcal{O}_2^{(k-n)}(x_2), \tag{28}$$

where $f_1$ is the fermion number of the operator $\mathcal{O}_1$. Without loss of generality, we work in an open ball ball $B^3 \subset \Sigma \times \mathbb{R}$ and consider the cycle

$$\mathfrak{C} := S_{x_2}^2 \times \{x_2\} \tag{29}$$

in the homology of the configuration space of two points in the ball, $H_\bullet(\mathcal{C}_2(B^3), \mathbb{Z})$. Here $S_{x_2}^2$ is a 2-sphere centred at $x_2$. We then define the secondary product as follows

$$\{[[\mathcal{O}_1]], [[\mathcal{O}_2]]\} := \left[\left[ \int_\mathfrak{C} (\mathcal{O}_1 \boxtimes \mathcal{O}_2)^{(2)} \right]\right]. \tag{30}$$

It is obvious that the definition of secondary product only depends on the cycle $\mathfrak{C}$ via its homology class in $H_2(\mathcal{C}_2(B^3), \mathbb{Z})$. In particular, we could have chosen the Hopf link instead of (29). More generally, any class in $H_2(\mathcal{C}_2(B^3), \mathbb{Z})$ gives a definition of secondary product that can be related to (30) up to an overall factor, as follows from

$$H_d(\mathcal{C}_2(B^3), \mathbb{Z}) := \begin{cases} \mathbb{Z}, & d = 0, \\ \mathbb{Z}, & d = 2, \\ 0, & \text{else}. \end{cases} \tag{31}$$

This fact can also be used to prove the first of the following two important relations:

$$\{[[\mathcal{O}_1]], [[\mathcal{O}_2]]\} = (-1)^{f_1 f_2 + 1} \{[[\mathcal{O}_2]], [[\mathcal{O}_1]]\}, \tag{32}$$

$$\{[[\mathcal{O}_1]], [[\mathcal{O}_2 \star \mathcal{O}_3]]\} = \{[[\mathcal{O}_1]], [[\mathcal{O}_2]]\} \star [[\mathcal{O}_3]] + (-1)^{f_1} [[\mathcal{O}_2]] \star \{[[\mathcal{O}_1]], [[\mathcal{O}_3]]\}, \tag{33}$$

$$\{[[\mathcal{O}_1]], \{[[\mathcal{O}_2]], [[\mathcal{O}_3]]\}\} = (-1)^{f_1 f_2} \{[[\mathcal{O}_2]], \{[[\mathcal{O}_1]], [[\mathcal{O}_3]]\}\} + \{\{[[\mathcal{O}_1]], [[\mathcal{O}_2]]\}, [[\mathcal{O}_2]]\}. \tag{34}$$

The second relation, which can be proven by considering the configuration space of three points, shows that the secondary product is a derivation of the algebra of local operators. The third is obviously the Jacobi identity.

## 2.4 SQM local operators and their commutator

We are ultimately interested in quantum mechanical systems defined along the real line $\mathbb{R}$. This is the supersymmetric quantum mechanics whose algebra was obtained in (7) and (18). Local operators in these quantum mechanical systems include operators that are local in 3d as well as their descendants along $\Sigma$. The aim of this section is to express the graded commutators between these operators in terms of the secondary product and the intersection pairing on $\Sigma$. This corresponds to the computation of the factorisation homology of the bulk algebra.

To this end, we denote the $\mathbb{R}$ component of a point $x \in \Sigma \times \mathbb{R}$ by $y$ (as before) and its components on the curve $\Sigma$ by $z$, so that $x = (y, z)$. The relevant cycles are then of the form (27)

$$\mathcal{C} \cong \mathcal{C}_\Sigma \times \{y\} \in H_\bullet(\Sigma, \mathbb{Z}) \times H_0(\mathbb{R}, \mathbb{Z}) \subset H_\bullet(\Sigma \times \mathbb{R}, \mathbb{Z}). \tag{35}$$

We claim that if two cycles $\mathcal{C}_\Sigma^1$ and $\mathcal{C}_\Sigma^2$ intersect transversally at a finite number of points, then for any two local operators[3] $\mathcal{O}_1, \mathcal{O}_2$

$$\left[\mathcal{O}_1\left(\mathcal{C}_\Sigma^1\right), \mathcal{O}_2\left(\mathcal{C}_\Sigma^2\right)\right] \cong \langle \mathcal{C}_\Sigma^1, \mathcal{C}_\Sigma^2 \rangle \{\mathcal{O}_1, \mathcal{O}_2\}. \tag{36}$$

Here $\langle\,,\,\rangle$ is the intersection pairing and $\{\,,\,\}$ is the secondary product. $[\,,\,]$ is the quantum-mechanical graded commutator. We also claim that if the cycles do not intersect at all, then the commutator simply vanishes.

We start by unpacking the quantum-mechanical commutator in terms of the 3d theory. To this end, consider the cycle $\mathfrak{C}_{12} \in H_\bullet(\mathcal{C}_2(\Sigma \times \mathbb{R}), \mathbb{Z})$ defined by a representative

$$\left(\mathcal{C}_\Sigma^1 \times \{y_1\}\right) \times \left(\mathcal{C}_\Sigma^2 \times \{y_2\}\right) \subset (\Sigma \times \mathbb{R}) \times (\Sigma \times \mathbb{R}), \tag{37}$$

with $y_1 < y_2$. Similarly, consider the cycle $\mathfrak{C}_{21}$ defined by the representative

$$\left(\mathcal{C}_\Sigma^1 \times \{y_2\}\right) \times \left(\mathcal{C}_\Sigma^2 \times \{y_1\}\right) \subset (\Sigma \times \mathbb{R}) \times (\Sigma \times \mathbb{R}). \tag{38}$$

The commutator is defined as the integration of the operator

$$(\mathcal{O}_1 \boxtimes \mathcal{O}_2)^{(k)} \tag{39}$$

over the cycle

$$\mathfrak{C} \cong \mathfrak{C}_{12} - \mathfrak{C}_{21}. \tag{40}$$

What we would like to show is that this cycle is homologous to the one defining the secondary product on the RHS of (36) when $\mathcal{C}_\Sigma^1$ and $\mathcal{C}_\Sigma^2$ intersect transversally, and that it vanishes when they do not intersect at all.

If the two cycles $\mathcal{C}_\Sigma^1$ and $\mathcal{C}_\Sigma^2$ do not intersect along the curve, then we are free to deform $\mathfrak{C}$ without changing its homology class so that $y_1 = y_2 = y$, $\mathfrak{C}_{12} = \mathfrak{C}_{21}$ and the cycle vanishes as claimed. Thus, any potential obstruction to the vanishing of the cycle has to come from intersections along the curve. Let us suppose then that the cycles intersect transversely at points

$$z_i \in \mathcal{C}_\Sigma^1 \cap \mathcal{C}_\Sigma^2, \tag{41}$$

and let $D_i \subset \Sigma$ be small disks surrounding them. In the complement of

$$\bigcup_i (D_i \times \mathbb{R}) \times (D_i \times \mathbb{R}) \subset (\Sigma \times \mathbb{R}) \times (\Sigma \times \mathbb{R}), \tag{42}$$

---

[3]We recall that differently from the previous section, we henceforth suppress the symbol $[[\ ]]$ denoting cohomology classes.

we can indeed set $y_2 = y_1$ in (37) and (38) without changing the homology class of $\mathcal{C}_{12}$ and $\mathcal{C}_{21}$. In any intersection with

$$(D_i \times \mathbb{R}) \times (D_i \times \mathbb{R}) \,, \tag{43}$$

since the intersection between $\mathcal{C}_\Sigma^1$ and $\mathcal{C}_\Sigma^1$ is transverse, we can modify (37) to

$$\epsilon_i \left( H_i^2 \right) \times \{(z_i, y)\} \,, \tag{44}$$

where $H_i^2$ is a lower hemisphere with centred at $(z_i, y)$ and the $\epsilon_i$ is a sign that depends on orientations. Doing the same with $\mathfrak{C}_{21}$, and noticing that contributions cancel out in the complement of (42), we obtain

$$\mathfrak{C} \cong \sum_{x_i} \epsilon_i S_{(z_i, y)}^2 \times \{(z_i, y)\} \,, \tag{45}$$

which corresponds to the desired cycle.

Finally, we would also need to compute the commutator of two operators descended along cycles whose interesection has dimension one or more. It follows from (31) that on dimensional grounds the commutator of such operators cannot be expressed in terms of the secondary product, and we believe that in fact it should vanish. This will be true in our examples.

In the following, we will use conventions for the cycles and their intersection pairing on $\Sigma$ that are summarised in appendix A. In particular,

$$\langle \Gamma_i, \Gamma_{g+j} \rangle = -\langle \Gamma_{g+j}, \Gamma_i \rangle = \delta_{i,j} \,, \tag{46}$$

where the first $g$ $\Gamma_i$'s are Poincaré dual to holomorphic one-forms. We also take a zero-cycle $\Lambda$ and a 2-cycle (abusing notation) $\Sigma$ so that

$$\langle \Lambda, \Sigma \rangle = 1 \,. \tag{47}$$

## 3 The free hypermultiplet in the B-twist

We now discuss the B-twisted free hypermultiplet. In flat space, this has fields

$$\{\phi^{Aa}\}_{a=1,2} \,, \tag{48}$$

with $A$ an $SU(2)_H$ index. The fields are subject to a reality condition

$$\left( \phi^{Aa} \right)^\dagger = \epsilon_{AB} \Omega_{ab} \phi^{Bb} \,, \tag{49}$$

where

$$\Omega_{ab} = \begin{pmatrix} 0 & 1 \\ -1 & 0 \end{pmatrix} \,. \tag{50}$$

The fermionic fields can be denoted by

$$\{\psi_\alpha^{\dot{A}a}\}_{a=1,2} \,, \tag{51}$$

where $\dot{A}$ is as usual an $SU(2)_C$ index. The supersymmetry transformations are schematically

$$Q_\alpha^{A\dot{A}} \phi^{Ba} = \epsilon^{AB} \psi_\alpha^{\dot{A}a} \,, \tag{52}$$

$$Q_\alpha^{A\dot{A}} \psi_\beta^{\dot{B}a} = i \epsilon^{\dot{A}\dot{B}} \partial_{\alpha\beta} \phi^{Aa} + \epsilon_{\alpha\beta} Z^{\dot{A}\dot{B}} \phi^{Aa} \,. \tag{53}$$

In the B-twist, all bosons remain scalars. It will be useful to denote

$$\phi^{1a} = X^a \,, \tag{54}$$

so that $(X^1, \bar{X}^2)$ and $(X^2, -\bar{X}^1)$ are doublets of $SU(2)_H$. The Fermions, however, decompose into scalars

$$\eta^a := -\delta^\alpha_{\dot{A}} \psi^{\dot{A}a}_\alpha \,, \tag{55}$$

and one forms

$$\chi^a_\mu := \frac{i}{2} \left(\sigma_\mu\right)^\alpha_{\dot{A}} \psi^{\dot{A}a}_\alpha \,. \tag{56}$$

Notice that with the definition in (23)

$$(X^a)^{(1)} = Q_\mu X^a = \chi^a_\mu \,. \tag{57}$$

In the absence of central charges, upon the identification

$$d\bar{X}^a := \eta^a \,, \tag{58}$$

the action of the supercharge $Q_{RW}$ can be interpreted as the action of the Dolbeault operator in a fixed complex structure. The Higgs branch chiral ring is by definition the ring of bosonic operators annihilated by $Q_{RW}$, that is the ring of holomorphic functions. Algebraically,

$$\mathbb{C}[M_H] := \mathbb{C}[X^1, X^2] \,, \tag{59}$$

where $M_H$ is the hyperkähler manifold

$$M_H := T^\vee \mathbb{C} \,. \tag{60}$$

The group $SU(2)_H$ acts on $M_H$ by rotating the complex structures.

There is a $U(1)$ flavour symmetry that rotates the fields $(X^1, X^2)$ by an opposite phase, and we therefore have a real mass parameter $m$ valued in the Lie algebra of $U(1)$. If $m \neq 0$, then $Z^{\dot{1}\dot{2}} = m \cdot J$ where $J$ is the generator of this symmetry. $Q_{RW}$ becomes a deformation of the Dolbeault operator

$$Q_{RW}(m) := e^{-\mu_{\mathbb{R},H} \cdot m} Q_{RW} e^{\mu_{\mathbb{R},H} \cdot m} \,, \tag{61}$$

where $\mu_{\mathbb{R},H}$ is the real moment map for the action of the flavour symmetry,

$$\mu_{\mathbb{R},H} = |X^1|^2 - |X^2|^2 \,. \tag{62}$$

## 3.1 Grading

Before moving on to the details of the Hilbert space, let us make a brief comment on the gradings of the fields. First, we grade the fields by the cohomological, or primary grading introduced above. This means that we grade the fields by their charge $F$ under the unbroken $U(1)_H \subset SU(2)_H$. Since $(X^1, \bar{X}^2)$, $(X^2, -\bar{X}^1)$ are doublets of $SU(2)_H$, the charges of the bosons under $U(1)_H$ are $\pm 1$, whereas the fermions are uncharged.

The cohomological grading can be thought of as a refinement of the fermion number, up to a caveat. In fact, the grading modulo two assigns grading 0 to fermions and grading 1 to bosons. This not what is expected from a fermion number, which usually assigns 0 to bosons and 1 to fermions. A $\mathbb{Z}$-grading that reduces to a usual fermion number modulo two can however be obtained by taking into consideration the flavour symmetry of the theory, as explained for example in [21].

We denote the charges of the fields under the flavour symmetry by $J$. The charges of all fields are equal to $J = \pm 1$. Thus, at the cost of replacing $U(1)_H$ with the diagonal subgroup

Table 1: Weights of hypermultiplet fields in the $B$-twist.

|   | $X^1$ | $X^2$ | $\chi_\mu^1$ | $\chi_\mu^2$ |
|---|---|---|---|---|
| $J$ | +1 | −1 | +1 | −1 |
| $F$ | 0 | +2 | −1 | +1 |
| $R$ | 0 | +1 | 0 | +1 |

$U(1)'_H \subset U(1)_H \times J_H$ in the definition of the primary grading, we obtain a cohomological grading that refines the fermion number. For notational simplicity, we keep symbols $F$ and $R$ for the primary and secondary grading defined with $U(1)'_H$ in place of $U(1)_H$. The resulting $J$, $F$ and $R$ charges are reported in Table 1. The algebraic syplectic form $\Omega_{ab}$ transforms with degrees $F = 2$, $R = 1$. Thus (remembering our convention for the primary grading (21)),

$$\mathbb{C}[M_H] \cong \mathrm{Sym}^\bullet[t\mathbb{C}[-2] \oplus \mathbb{C}]. \tag{63}$$

## 3.2 Effective quantum mechanics

In [11] the free hypermultiplet B-twisted on $\mathbb{R} \times \Sigma$ was studied from the point of view of an effective supersymmetric quantum mechanics on $\mathbb{R}$ in the presence of real mass deformations, and the vector space of supersymmetric ground states was constructed. The result can be understood in terms of the Rozansky-Witten invariants of $M_H$ [27].[4] In this section, we revisit the construction and manifest the structure of the vector space as a module for the algebra of local operators in the SQM.

### 3.2.1 Vector space of supersymmetric ground states

The effective supersymmetric quantum mechanics of [11] was derived by considering a particular localisation scheme. The scheme requires the path integral to localise on configurations holomorphic on $\Sigma$. The result of the procedure is a supersymmetric quantum mechanics with the following $(0,4)$ multiplets:

- A 1d hypermultiplet valued in $M_H$, which corresponds to the coefficients in the expansion of $(X^a, \eta^a)$ in terms of a basis for $H^0(\Sigma, \mathcal{O})$;

- $g$ Fermi multiplets $\chi_i^a$ valued in $M_H[1]$ arising from the modes of the one-form fermions $\chi_\mu^a$ on $\Sigma$. More precisely, the Fermi multiplets are the $g$ coefficients in the expansion of $\chi_\mu^a$ in terms of a basis of holomorphic one-forms $H^1(\Sigma, \mathcal{O})$, $w_\mu^i$

$$\chi_\mu^a = \sum_{i=1}^{g} \chi_i^a w_\mu^i. \tag{64}$$

As explained in [11], the quantum mechanics can be viewed as a standard $(0,4)$ quantum mechanics with target $M_H = T^\vee \mathbb{C}$ and endowed with a hyper-holomorphic vector bundle $\mathcal{F} = T_{M_H} \otimes \mathbb{C}^g$. Notice that if one inserts background connections for the flavour symmetry, the number of fluctuations changes (see appendix C). From the quantum mechanical perspective, this induces a modified vector bundle on the target [15].

As usual in quantum mechanics, the Hilbert space was defined to be the space of square-integrable functions on the target. The space of supersymmetric ground states is then the

---

[4]The derivation of the space of states assigned to a Riemann surface in the absence of real masses can also be approached from the point of view of the twisted formalism [28]. See also [29]).

subspace of the Hilbert space annihilated by all four supercharges. Upon turning on real a mass, the $Q_{RW}$ supercharge is deformed according to (61)

$$Q_{RW}(m) := e^{-\mu_{\mathbb{R},H} \cdot m} Q_{RW} e^{\mu_{\mathbb{R},H} \cdot m}, \tag{65}$$

with similar expressions for the other supercharges. The following square-integrable wave-functions arising from the 1d hypermulitplet and annihilated by all supercharges were found[5]

$$\left(X^1\right)^{k_1} \left(\bar{X}^2\right)^{k_2} e^{-m\left(|X^1|^2 + |X^2|^2\right)} d\bar{X}^2. \tag{66}$$

Accounting for the Fermi multiplets, we have

$$\mathcal{H}_{m>0} = \widehat{\mathrm{Sym}^\bullet V}, \tag{67}$$

where

$$V = \xi(\mathbb{C} \oplus t^{-1}\mathbb{C}[2] \oplus t^{-1}\mathbb{C}^g[1] \oplus \mathbb{C}^g[1]). \tag{68}$$

Here $\xi$ is a grading parameter that keeps track of the flavour symmetry, and the hat represents multiplication by the square-root of the determinant of $V$. $t^{-1}\mathbb{C}[2]$ corresponds to the grading of the operator $\bar{X}^2$.

Having established what the vector space of supersymmetric ground states is, the next step is to determine the quantum algebra of observables that acts on it.

### 3.2.2 The algebra of observables

A key fact that we need and that was derived in [21] is the following:

$$\{X^1, X^2\} = 1. \tag{69}$$

The derivation is based on the following simple computation

$$d(X^a)^{(2)} = \Omega^{ab} d \star d\left(\bar{X}^b\right) = \Omega^{ab} \frac{\delta S}{\delta X^b}, \tag{70}$$

and on the fact that in the path integral $\frac{\delta S}{\delta X^b}(x) X^a(y) \sim \delta^a_b \delta(x-y)$. It then follows from (36) and Stokes' theorem that

$$\left[X^1(x), X^2(\Sigma)\right] = 1, \tag{71}$$

as well as

$$\left[X^1(\Gamma_i), X^2(\Gamma_{j+g})\right] = \delta_{ij}, \tag{72}$$

which endows the space of local operators with the structure of Heisenberg algebras. Notice that in (72) the commutator is graded, that is, it is actually an anti-commutator.

### 3.2.3 Vector space of supersymmetric ground states as a module

Let us fix $m > 0$. The case $m < 0$ is analogous. We can represent states in the vector space as follows:

$$|k_1, k_2, f_1, \cdots, f_{2g}\rangle, \tag{73}$$

where $k_1, k_2 \in \mathbb{N}$ represent the powers of the bosonic Fock spaces whereas $f_i \in \mathbb{Z}_2$ the powers of the fermionic ones. In particular, the vector subspace spanned by the above state is

$$\xi^{k_1 + k_2 + \sum_{i=1}^{2g} f_i} t^{-k_2 - \sum_{i=1}^{g} f_i} \mathbb{C}\left[-2k_2 - \sum_{i=1}^{2g} f_i\right]. \tag{74}$$

---

[5]Notice that if we remove the Gaussian measure by absorbing it into the operators and use an appropriate normalisation, in the $m \to \infty$ limit the wave-functions simply become polynomials in $X^1$ and derivatives of delta-functions around $X^2$ [30].

We can now derive the action of the generators $X^1$ and $X^2$ of the Higgs branch chiral ring $X^1$ and $X^2$ and their descendants. If we picture a state in terms of a field configuration on $\mathbb{R}^+ \times \Sigma$, the action literally corresponds to bringing the operator in question to $y \to 0$.

Now, it is obvious that

$$X^1 \cdot |k_1, k_2, f_1, \cdots, f_{2g}\rangle = |k_1 + 1, k_2, f_1, \cdots, f_{2g}\rangle. \tag{75}$$

The action $X^2$ can be computed by means of the following observation. Since

$$Q_{RW}(m)\left(\left(X^1\right)^{k_1}\left(\bar{X}^2\right)^{k_2} e^{-m\left(|X^1|^2+|X^2|^2\right)}\right) = k_2 \left(X^1\right)^{k_1}\left(\bar{X}^2\right)^{k_2-1} e^{-m\left(|X^1|^2+|X^2|^2\right)} d\bar{X}^2 \tag{76}$$

$$- 2m X^2 \left(X^1\right)^{k_1}\left(\bar{X}^2\right)^{k_2} e^{-m\left(|X^1|^2+|X^2|^2\right)} d\bar{X}^2, \tag{77}$$

the two terms on the RHS must be cohomologous. Thus,

$$X^2 \cdot |k_1, k_2, f_1, \cdots, f_{2g}\rangle = \frac{-k_2}{2m} |k_1, k_2 - 1, f_1, \cdots, f_{2g}\rangle. \tag{78}$$

It then follows from (71) and its conjugate that

$$X^1(\Sigma) \cdot |k_1, k_2, f_1, \cdots, f_{2g}\rangle = -2m|k_1, k_2 + 1, f_1, \cdots, f_{2g}\rangle, \tag{79}$$

$$X^2(\Sigma) \cdot |k_1, k_2, f_1, \cdots, f_{2g}\rangle = -k_1|k_1 - 1, k_2, f_1, \cdots, f_{2g}\rangle. \tag{80}$$

The action of the fermionic operators is straightforward. Recall that we have selected a basis of one-cycles so that $\gamma_i$ for $i \in \{1, \ldots, g\}$ are dual to holomorphic one-forms. Thus by means of (57) we see that $X^a(\gamma_i)$ for $i \in \{1, \ldots g\}$ correspond to the creation operators. By (72), $X^a(\gamma_{g+i})$ for $i \in \{1, \ldots g\}$ can then be identified with the fermionic annihilation operators.

Finally, it is interesting to couple the system to a background flat connection for the flavour symmetry. This will in general change the number of Heisenberg algebras, as we briefly mention in appendix C.

# 4 SQED[1] in the A-twist

We now discuss the A-twist of SQED[1], the theory of a gauged hypermultiplet of gauge charge 1. Besides the hypermultiplet fields

$$\left(\phi^{Aa}, \psi_\alpha^{\dot{A}a}\right), \tag{81}$$

defined above, we have a vectormultiplet with components

$$\left(A_\mu, \sigma^{\dot{A}\dot{B}}, \lambda_\alpha^{A\dot{A}}, D^{AB}\right), \tag{82}$$

that are the gauge connection, scalars, gauginos, and auxiliary fields respectively. In addition, there is a scalar field $\gamma$ dual to the field strength $F_A$, satisfying

$$d\gamma = \star_{3d} F_A, \tag{83}$$

where $\star_{3d}$ is the Hodge star operator in three dimensions. The theory enjoys a $U(1)$ topological symmetry that rotates the dual photon, with current $F_A$. If we denote the conserved charge by $J$, then we have

$$Z^{12} = \zeta \cdot J, \tag{84}$$

where $\zeta$ is a real FI parameter.

In the A-twist, the bosonic fields of the hypermultiplet $\phi^{Aa}$ become spinors that for simplicity we will still denote by

$$X^a := \phi^{1a}\,,\tag{85}$$

suppressing the spinor index. The gauginos transform either as scalars or one-forms of the improved Lorentz group

$$\lambda^{\dot A}_\mu := \frac{i}{2}\left(\sigma_\mu\right)^\alpha_A \lambda^{A\dot A}_\alpha\,,\tag{86}$$

$$\lambda^{\dot A} := -\delta^\alpha_A \lambda^{A\dot A}_\alpha\,.\tag{87}$$

## 4.1 The Coulomb branch and mirror symmetry

The A-twist preserves the Coulomb branch chiral ring, which is generated by two monopole operators $v_+$ and $v_-$ and also includes the complex scalar

$$\varphi := \sigma^{\dot 1 1} + i\sigma^{\dot 2 2}\,.\tag{88}$$

The monopole operators can be understood semi-classically in terms of the dual photon as a path-integral insertion of the operator

$$v_\pm = e^{\pm(\sigma + i\gamma)}\,,\tag{89}$$

where

$$\sigma := \sigma^{\dot 1 \dot 2}\,.\tag{90}$$

Notice that by writing this we have implicitly chosen a complex structure on the Coulomb branch, namely the one that is also used to define $Q_{mRW}$. The effect of the insertion of the monopole operator $v_\pm$ is to require that the first Chern class of the gauge bundle around the point of insertion is $\pm 1$.

The algebra of monopole operators was computed in [31] (see also [32]), and reads

$$v_- v_+ = \varphi\,.\tag{91}$$

In particular, it allows to express $\varphi$ in terms of $v_\pm$. Thus, we have

$$\mathbb{C}[M_C] \cong \mathbb{C}[v_+, v_-]\,,\tag{92}$$

the ring of polynomial functions in two variables generated by $v_+$ and $v_-$. A mathematical, algebraic definition of the Coulomb branch chiral ring was first proposed in [33], refined in [34] and reviewed at an introductory level for example in [35]. We review some elementary aspects of this definition in appendix D.

As in the B-twist, we are interested in the secondary product between local operators of the Coulomb branch chiral ring. One way to compute it is simply to exploit the mirror map. In fact, the theory is mirror dual to a free twisted hypermultiplet that relates the local operators as follows [36]

$$\begin{pmatrix} X^1 \\ X^2 \\ X^1 X^2 \end{pmatrix} \leftrightarrow \begin{pmatrix} v_+ \\ v_- \\ \varphi \end{pmatrix}\,.\tag{93}$$

Since the secondary product is scale-independent, this implies

$$\{\varphi, v_\pm\} = \pm v_\pm\,,\tag{94}$$

$$\{v_+, v_-\} = 1\,.\tag{95}$$

It is instructive to explain how this can be computed via first principles. First, notice that the second line can be inferred from the first by means of the monopole algebra (91) together with the derivation identity (33). The first can be computed following [21] by noticing that the second descendant of $\varphi$ is $\frac{1}{4\pi}(F_A + \star_{3d}D\sigma)$.[6] Integrated over a sphere surrounding $v_\pm$, by definition of the monopole operators this gives $\pm 1$. As $\varphi$ can be interpreted as the complex moment map for the topological symmetry, this equation can in fact be read as the statement that the monopole operators have integer charges $\pm 1$. In the next sections we will identify the descendants mirror to (57) and we will provide an operational definition for them.

Finally, as for the mirror B-twist of the free hypermultiplet, we would like to define a cohomological grading that agrees with the standard fermionic grading. As in section 3.1 and as already done in [11], we need to mix the R-symmetry grading with the topological symmetry to obtain

Table 2: Weights of monopole operators in the SQED[1] A-twist.

|     | $v_+$ | $v_-$ | $\varphi$ |
| --- | --- | --- | --- |
| $J$ | $+1$ | $-1$ | $0$ |
| $F$ | $0$ | $+2$ | $+2$ |
| $R$ | $0$ | $+1$ | $+1$ |

## 4.2 Effective quantum mechanics

Let us now turn to the Hilbert space of SQED[1]. We first review the construction [11] and then study the action of the monopole operators and their descendants this set-up. As expected from the mirror map, the Hilbert space will turn out to be a Fock space for the Heisenberg algebras generated by the monopoles and their descendants.

The Hilbert space was constructed in [11] as follows. The theory can be recast in terms of a Landau-Ginzburg quantum mechanics with Kähler target given by the Kähler quotient of the space of smooth fields configurations on $\Sigma$ by the action of the gauge group. The gauge group has moment map

$$\star_{2d}F_A + \mu_{\mathbb{R}}, \tag{96}$$

and so the relevant equations are

$$\star_{2d}F_A + \mu_{\mathbb{R}} = \zeta. \tag{97}$$

The quantum mechanics is endowed with a superpotential on the target

$$W = \int_\Sigma X^1 \bar{\partial}_A X^2, \tag{98}$$

which imposes the complex moment map equation for the gauge symmetry as well as the kinetic equations for the fields $X^\alpha$ on the curve $\Sigma$.[7] This superpotential defines a critical locus, which is $(-1)$-shifted symplectic with respect to the $F$ grading. Passing to a finite-dimensional algebraic model (imposing holomorphicity of the fields), the $(-1)$-shifted symplectic structure allows for a geometric quantisation of the quantum mechanics. In the simple situation where the target space of the finite-dimensional model is actually smooth, as the present one turns

---

[6]Here $\star_{3d}$ is the Hodge operator in 3d. With respect to [21] we are re-absorbing a factor of $1/2\pi$ in $\varphi$ for the sake of compatibility with the mirror map (93).

[7]Notice that due to the twist, $X^1$ and $X^2$ are valued in $\Omega^0(\Sigma, E \otimes K_\Sigma^{1/2})$ and $\Omega^0(\Sigma, E^{-1} \otimes K_\Sigma^{1/2})$ respectively.

out to be, the quantisation recovers (up to shifts in the gradings) its de Rham cohomology [11]. Requiring $\zeta \neq 0$, the equations defining the target space are

$$\star_{2d}F_A + e^2\left|X^1\right|^2 = \zeta\,,\ \bar{\partial}_A X^1 = 0,\ X^2 = 0,\quad \zeta > 0\,, \tag{99}$$

$$\star_{2d}F_A - e^2\left|X^2\right|^2 = \zeta\,,\ \bar{\partial}_A X^2 = 0,\ X^1 = 0,\quad \zeta < 0\,, \tag{100}$$

where we have already used the complex moment map $X^1 X^2 = 0$ and the fact that both $X^1$ and $X^2$ must be holomorphic. Here $\star_2$ is the Hodge star on the Riemann surface.

Since the cases $\zeta > 0$ and $\zeta < 0$ are similar, we restrict our discussion to $\zeta > 0$. It is well-known that solutions to the above equation can be parametrized by pairs $(E, X^1)$ consisting of a holomorphic line bundle $E$, with holomorphic structure induced by the $(0,1)$ component of $A$, and a holomorphic section $X^1$ of $E \otimes K_\Sigma^{1/2}$ that is not vanishing. The solution space decomposes into disjoint unions with components labelled by the degree $d$ of the line bundle $E$. If we take the limit $\zeta/\text{Vol}(\Sigma)d \to \infty$ for each $d$, then it is also well-known that pairs $(E, X^1)$ are parametrized by the locations of the zeros of $X^1$. Physically, the zeros can be interpreted as the centres of the vortices. In this limit, an infinite number of vortices is allowed.

Thus, fixing a topological degree $d$, we can identify the moduli space of solutions $M_d$ with

$$M_d = \Sigma(n)\,,\quad n = d + g - 1\,, \tag{101}$$

where $\Sigma(n)$ is the $n$-fold symmetric product of the curve $\Sigma$. The moduli space space for each $d$ is indeed a smooth algebraic variety, and the Hilbert space is therefore de Rham cohomology of the disjoint union of symmetric products for $n \geq 0$,

$$\mathcal{H} \cong \bigoplus_{n \geq 0} H^\bullet(\Sigma(n))\,, \tag{102}$$

where $\bullet$ corresponds to the primary $R$-grading. Keeping track of all the gradings, as a graded vector space this can be repackaged into the expression

$$\mathcal{H} = \widehat{\text{Sym}^\bullet V}\,, \tag{103}$$

for

$$V = \xi(\mathbb{C} \oplus t^{-1}\mathbb{C}^g[1] \oplus \mathbb{C}^g[1] \oplus t^{-1}\mathbb{C}[2])\,. \tag{104}$$

Here $\xi$ is a weight that keeps track of the degree $d$. In the next sections we will interpret this as a Fock space for the Heisenberg algebras generated by the monopole operators and their descendants. In particular, the operator $v_+$ will turn out to be a creation operator for the $\mathbb{C}$ component whereas $v_-(\Sigma)$ is a creation operator for the $t^{-1}\mathbb{C}[2]$ component.

### 4.2.1 Algebra of observables

As in the B-twist, we would like to compute the quantum-mechanical commutator between the monopole operators and their descendants. The commutator can be expressed in terms of the secondary product as in (36). We have computed the secondary product in (95)

$$\{v_+, v_-\} = 1\,. \tag{105}$$

Therefore by (36),

$$[v_+, v_-(\Sigma)] = 1\,, \tag{106}$$

with similar expressions for the first descendants

$$\left[v_+(\Gamma_i), v_-(\Gamma_{g+i})\right] = 1\,. \tag{107}$$

Note that these descendants are mirror to the descendants introduced in (57) in the B-twist.

### 4.2.2  Vector space of supersymmetric ground states as a module

Let us consider the monopole operators $v_+(x)$ and $v_-(x)$ inserted at a point $x \in \Sigma \times \mathbb{R}$ as well as their descendants. Recall that in non-supersymmetric 3d gauge theories $v_\pm(x)$ are defined, at the level of the path integral, by the following procedure:

- Remove $x$ from space-time and perform the path integral by integrating over gauge bundles whose first Chern class evaluated on a small sphere surrounding $x$ is equal to $\pm 1$.

Since it is known what constraints do the monopole operators impose on the gauge fields, one can derive how they act on solutions to the BPS equations. A similar strategy was adopted in [37] on the set-up $\mathbb{R}^2_\epsilon \times \mathbb{R}$, where the BPS equations were solved by vortex configurations on $\mathbb{R}^2_\epsilon$ with a prescribed behaviour at infinity.

In mathematical terms, monopole operators were identified with Hecke correspondences between vortex moduli spaces, which as expected from the procedure highlighted above increase and decrease the degree of the gauge bundle by one. In more physical terms, they are operators that create and destroy vortices.

The same arguments can be applied to our set-up $\Sigma \times \mathbb{R}$. Thus, our starting point is that the monopole operator $v_+(x)$, $x = (y, z)$ creates a vortex at $z \in \Sigma$, and the monopole operator $v_-(x)$ destroys a vortex at $z \in \Sigma$.

As we quickly reviewed above, the moduli space of pairs $(E, X^1)$ with $E$ a holomorphic line bundle of degree $d \in \mathbb{Z}$, and $X^1$ a holomorphic section of $E \otimes K_\Sigma^{1/2}$, can be parametrized by the $d + g - 1$ zeros of $X$. Physically, these zeros can be thought of as the centres of the vortices. Let us interpret $\mathbf{p} \in \Sigma(n)$ as a positive divisor on $\Sigma$,

$$\mathbf{p} = z_1 + z_2 + \cdots + z_n, \tag{108}$$

with positive coefficients. Operations of creation and annihilation of vortices at a point $\{z\}$ can be described by first defining cycles

$$E_n^{\{z\}} := \{(\mathbf{p}, \mathbf{q}) \subset \Sigma(n-1) \times \Sigma(n) \mid \mathbf{q} - \mathbf{p} = z\}, \tag{109}$$

$$F_n^{\{z\}} := \{(\mathbf{p}, \mathbf{q}) \subset \Sigma(n+1) \times \Sigma(n) \mid \mathbf{p} - \mathbf{q} = z\}. \tag{110}$$

These cycles are the starting point of the needed correspondences between moduli spaces. We can construct actions on the Hilbert space concretely by noticing that these cycles induce classes

$$\mathcal{E}_n^{\{z\}} \in \bigoplus_{k,l} H^{2(n-1)-l}(\Sigma(n-1)) \otimes H_k(\Sigma(n)), \tag{111}$$

$$\mathcal{F}_n^{\{z\}} \in \bigoplus_{k,l} H^{2(n+1)-l}(\Sigma(n+1)) \otimes H_k(\Sigma(n)), \tag{112}$$

where we utilised the Künneth formula and Poincaré duality. Thus, it is natural to identify

$$v_+(z) \cdot \cong \sum_{n>0} \mathcal{E}_n^{\{z\}} \cdot, \tag{113}$$

$$v_-(z) \cdot \cong \sum_{n \geq 0} \mathcal{F}_n^{\{z\}} \cdot, \tag{114}$$

where the action $\cdot$ on the RHS is given by pairing cohomology with homology.

The above operators are first analogues of the Nakajima operators [24]. Nakajima operators can be defined for any cycle $\mathcal{C}_\Sigma$ of the Riemann surface, by generalising (109) to

$$E_n^{\mathcal{C}_\Sigma} := \{(\mathbf{p}, \mathbf{q}) \subset \Sigma(n-1) \times \Sigma(n) \mid \mathbf{q} - \mathbf{p} = z \in \mathcal{C}_\Sigma\}, \tag{115}$$

$$F_n^{\mathcal{D}_\Sigma} := \{(\mathbf{p}, \mathbf{q}) \subset \Sigma(n+1) \times \Sigma(n) \mid \mathbf{p} - \mathbf{q} = z \in \mathcal{D}_\Sigma\}, \tag{116}$$

and by using as above the Künneth decomposition and Poincaré duality. The resulting operators are obvious candidates for the descendants of $v_+$ and $v_-$ along the respective cycles. In fact, the semi-classical description (89) together with[8]

$$(i\gamma + \sigma)^{(1)} = \lambda_\mu^{\dot{2}}, \tag{117}$$

suggests that a descendant of a monopole operator essentially corresponds to a collection of monopole operators dressed by gauginos and distributed along the prescribed cycle. Formally,

$$v_\pm(\gamma) = \int_\gamma v_\pm \lambda_\mu^{\dot{2}} dx^\mu. \tag{118}$$

We claim that this is consistent with (115).

To check that our physical realisation is correct, we have to make sure that the representation on the Hilbert space of monopole operators and their descendants satisfies the correct algebraic relations. We have encountered two kinds of relations, namely the relations (91) and the Heisenberg algebra relations (106) and (107). The Heisenberg algebra relations can be proven along the same lines of the original proof of Nakajima in the case of Hilbert scheme of points on surfaces [24]. The adaptation of the proof to this case is sketched in appendix B. As for the others, consider

$$v_+(z) \cdot v_-(z). \tag{119}$$

Our definitions above are equivalent to the following

$$v_+(z) \cdot v_-(z) = j_{z*} j_z^*, \tag{120}$$

where $j_z$ is the map that adds the point $z$ to a divisor

$$j_z : z_1 + z_2 + \ldots + z_n \mapsto z_1 + z_2 + \ldots + z_n + z. \tag{121}$$

Now the class of $j_z(\Sigma(n))$ in $H^{1,1}(\Sigma(n+1), \mathbb{C})$ is the generator of this cohomology group. We call this $\eta_n$.[9] Thus, by the projection formula, for any class $\alpha$ (see e.g. [38])

$$j_{z*} j_z^* \alpha = \eta_n \wedge \alpha, \tag{122}$$

where $j_*$ is the push-forward in cohomology induced by $j$ and $j^*$ is the pull-back. This is consistent with the expected action of $\varphi$, which is essentially dictated by the cohomological gradings [37]. In fact, the cohomological charge of $\varphi$ is 2, and since the operator acts by multiplication the only possible operation is wedging by a form of the same cohomological degree. Such a form is represented by $\eta_n$.

## 5 Summary and future directions

In this paper, we have explicitly constructed the action of local operators on the space of supersymmetric ground states in the effective quantum mechanics obtained obtained from a 3d theory twisted on the geometry $\Sigma \times \mathbb{R}$. Although we have focussed on the simplest mirror pair, where the geometric interpretation of the action is cleanest, the above results can be generalised to the broad class of theories studied in [11]. There are several interesting directions that would be worthwhile pursuing, for instance:

---

[8]To compute this, note that

$$d\gamma^{(1)} = (d\gamma)^{(1)} = d \star_{3d} A_\mu^{(1)}.$$

[9]See A for our conventions.

- These concrete geometric constructions should constitute an aspect of the interesting recent works [39, 40] on shifted quiver algebras and BPS crystals;

- The most natural next step would be to insert background connections, line defects as well as boundary conditions in this set-up;

- Related to the last point, it would be interesting to study the effect of the insertion of deformed $(0, 4)$ boundary conditions that lead to Vertex Operator Algebras, with the aim of making contact with the mathematical work [4] and eventually with the Geometric Langlands Program;

- Finally, it would be interesting to explore similar phenomena in higher dimensions.

## Acknowledgements

The idea of writing this paper originated in the context of a Visiting Graduate Fellowship at the Perimeter Institute for Theoretical Physics in 2019, and we gratefully acknowledge discussions with A. Braverman and D. Gaiotto that led us to think about the problem. We thank M. Bullimore and T. Dimofte for several useful discussions during the development of the paper. Finally, we would like to thank anonymous referees for useful comments.

## A   Homology and Cohomology of $\mathrm{Sym}^d(\Sigma)$

We start with $\mathrm{Sym}^1(\Sigma) = \Sigma$. We fix once and for all a complex structure on $\Sigma$ as well as a canonical basis $a_1, \cdots, a_g, b_1, \cdots, b_g$ of $H_1(\Sigma, \mathbb{C})$. Then, we pick basis elements

$$\alpha \in H^0(\Sigma, \mathbb{C}) \cong \mathbb{C}, \tag{A.1}$$

$$\gamma_1, \cdots, \gamma_{2g} \in H^1(\Sigma, \mathbb{C}) \cong \mathbb{C}^{2g}, \tag{A.2}$$

$$\eta \in H^2(\Sigma, \mathbb{C}) \cong \mathbb{C}, \tag{A.3}$$

that satisfy the following conditions. For $1 \le i \le g$, $\gamma_i \in H^{1,0}(\Sigma, \mathbb{C})$ (with respect to the chosen complex structure) and if we denote the intersection pairing by $\langle\ ,\ \rangle$,

$$\langle \gamma_i, \gamma_{g+j} \rangle = -\langle \gamma_{g+j}, \gamma_i \rangle = \delta_{i,j}, \tag{A.4}$$

for $i, j \le g$. This is possible since as it is well-known, there is a basis $w_i$ for $H^{1,0}(\Sigma, \mathbb{C})$ such that

$$\int_{a_i} w_j = \delta_{ij}, \tag{A.5}$$

and so we can simply take Poincaré duals of $a_i$ to complete the basis of $H^1(\Sigma, \mathbb{C})$. Then, we fix a basis of cycles for the homology groups $H_i(\Sigma, \mathbb{C})$ that is dual to the basis above. With a slight abuse of notation we pick basis elements

$$\Lambda \in H_0(\Sigma, \mathbb{C}) \cong \mathbb{C}, \tag{A.6}$$

$$\Gamma_1, \ldots, \Gamma_{2g} \in H_1(\Sigma, \mathbb{C}) \cong \mathbb{C}^{2g}, \tag{A.7}$$

$$\Sigma \in H_2(\Sigma, \mathbb{C}) \cong \mathbb{C}, \tag{A.8}$$

so that

$$(\Sigma, \alpha) = 1, \quad (\Gamma_i, \gamma_i) = 1, \quad (\Lambda, \eta) = 1, \tag{A.9}$$

where $(\cdot, \cdot)$ is the dual pairing between homology and cohomology. Below we will make use of two orderings of the basis

$$\mathcal{C}_\Sigma^a \in \{\Lambda, \Gamma_1, \cdots \Gamma_{2g}, \Sigma\},\, a \in \{1, \cdots, 2g+2\}, \tag{A.10}$$

$$\mathcal{D}_\Sigma^a \in \{\Sigma, \Gamma_{g+1}, \cdots, \Gamma_{2g}, \cdots, \Gamma_1, \cdots, \Gamma_g, \Lambda\},\, a \in \{1, \cdots, 2g+2\}. \tag{A.11}$$

In order to determine our conventions for the homology and cohomology of $\mathrm{Sym}^d(\Sigma)$, we make use of the identity

$$H^\bullet(\mathrm{Sym}^d(\Sigma), \mathbb{C}) \cong H^\bullet(\Sigma^d, \mathbb{C})^{S_d}, \tag{A.12}$$

where $S_d$ is the permutation group. The right-hand side consists of permutation-invariant elements in the cohomology of the $d$-fold product of $\Sigma$. Thus, let us introduce

$$\gamma_{i,j} = 1 \otimes \cdots \otimes 1 \otimes \gamma_i \otimes 1 \otimes \cdots \otimes 1 \in H^1(\Sigma^d, \mathbb{C}), \tag{A.13}$$

$$\eta_j = 1 \otimes \cdots \otimes 1 \otimes \eta \otimes 1 \otimes \cdots \otimes 1 \in H^2(\Sigma^d, \mathbb{C}), \tag{A.14}$$

where the generator appears in the $j$-th factor. The classes

$$\widetilde{\gamma}^i = \sum_{j=1}^d \gamma_{i,j}, \quad \widetilde{\eta} = \sum_{j=1}^d \eta_j, \tag{A.15}$$

then descend to $H^\bullet(\Sigma^d, \mathbb{C})^{S_d}$, and in fact generate it.

# B  Nakajima relations

Let us define the cycles

$$E_n^a := \{(\mathbf{p}, \mathbf{q}) \subset \Sigma(n-1) \times \Sigma(n) \mid \mathbf{q} - \mathbf{p} = z \in \mathcal{C}_\Sigma^a\}, \tag{B.1}$$

$$F_n^a := \{(\mathbf{p}, \mathbf{q}) \subset \Sigma(n+1) \times \Sigma(n) \mid \mathbf{p} - \mathbf{q} = z \in \mathcal{D}_\Sigma^a\}, \tag{B.2}$$

where the $\mathcal{C}_\Sigma^a$'s and $\mathcal{D}_\Sigma^a$'s denote the homology basis we chose above. These cycles induce classes

$$\mathcal{E}_n^a \in \bigoplus_{k,l} H^{2n-2-l}(\Sigma(n-1)) \otimes H_k(\Sigma(n)), \tag{B.3}$$

$$\mathcal{F}_n^a \in \bigoplus_{k,l} H^{2n+2-l}(\Sigma(n+1)) \otimes H_k(\Sigma(n)), \tag{B.4}$$

where we used the Künneth formula and Poincaré duality. By means of the dual pairing $(\cdot, \cdot)$ we can define a convolution product between these operators, which we will denote by $\cdot$, as well as an action on

$$\mathcal{H} = \bigoplus_{n \geq 0} H^{\bullet, \bullet}(\Sigma(n), \mathbb{C}). \tag{B.5}$$

## B.1  Heisenberg algebra relations

We would like to check that the above operators (111) (112) satisfy the Heisenberg algebra

$$\mathcal{E}_{n-1}^a \cdot \mathcal{E}_n^b - (-1)^{\dim_\mathbb{R}(\mathcal{C}_\Sigma^a)\dim_\mathbb{R}(\mathcal{C}_\Sigma^b)} \mathcal{E}_{n-1}^b \cdot \mathcal{E}_n^a = 0, \tag{B.6}$$

$$\mathcal{F}_{n+1}^a \cdot \mathcal{F}_n^b - (-1)^{\dim_\mathbb{R}(\mathcal{D}_\Sigma^a)\dim_\mathbb{R}(\mathcal{D}_\Sigma^a)} \mathcal{F}_{n+1}^b \cdot \mathcal{F}_n^a = 0, \tag{B.7}$$

$$\mathcal{E}_{n+1}^a \cdot \mathcal{F}_n^b - (-1)^{\dim_\mathbb{R}(\mathcal{C}_\Sigma^a)\dim_\mathbb{R}(\mathcal{D}_\Sigma^b)} \mathcal{F}_{n-1}^b \cdot \mathcal{E}_n^a = \delta_{ab} c[\Delta(n)], \tag{B.8}$$

where $\Delta(n)$ is the diagonal in $\Sigma(n) \times \Sigma(n)$ and $c$ is a constant. We focus on (B.8), which is a little more subtle. The other relations are similar. Consider the spaces

$$M := \Sigma(n) \times \Sigma(n+1) \times \Sigma(n), \tag{B.9}$$

$$M' := \Sigma(n) \times \Sigma(n-1) \times \Sigma(n). \tag{B.10}$$

Consider the first term on the LHS of (B.8). We think of points in $\Sigma(n)$, $\Sigma(n-1)$ and $\Sigma(n+1)$ as divisors and define $N$ to be the set of triples

$$(\mathbf{p}, \mathbf{q}, \mathbf{r}) \subset M \tag{B.11}$$

satisfying $\mathbf{q} - \mathbf{p} = z$, $\mathbf{q} - \mathbf{r} = w$ for some $z \in \mathcal{C}_\Sigma^a$, $w \in \mathcal{D}_\Sigma^b$. The resulting operator can then be obtained as the class of

$$\pi_{13}(N), \tag{B.12}$$

with an obvious notation for the projection into the first and third factor. Similarly, we define the set of triples $N' \subset M'$

$$(\mathbf{p}', \mathbf{q}', \mathbf{r}') \subset M' \tag{B.13}$$

satisfying $\mathbf{p}' - \mathbf{q}' = w'$, $\mathbf{r}' - \mathbf{q}' = z'$ for some $z' \in \mathcal{C}_\Sigma^a$, $w' \in \mathcal{D}_\Sigma^a$. The second operator on the LHS of (B.8) can then be obtained as the class of

$$\pi_{13}(N'). \tag{B.14}$$

If $z \neq w$, $z' \neq w'$ then we can define explicit maps

$$\mu : N \to N', \tag{B.15}$$

$$\mu((\mathbf{p}, \mathbf{q}, \mathbf{r})) = (\mathbf{p}, \mathbf{p} \cap \mathbf{r}, \mathbf{r}), \tag{B.16}$$

$$\nu : N' \to N, \tag{B.17}$$

$$\nu((\mathbf{p}', \mathbf{q}', \mathbf{r}')) = (\mathbf{p}', \mathbf{p}' + \mathbf{r}' - \mathbf{p}' \cap \mathbf{r}', \mathbf{r}'). \tag{B.18}$$

These maps are clearly inverse to each other, and provide an isomorphism between the two sets of triples

$$U = \{(\mathbf{p}, \mathbf{q}, \mathbf{r}) \subset N \,|\, z \neq w\}, \tag{B.19}$$

$$U' = \{(\mathbf{p}', \mathbf{q}', \mathbf{r}') \subset N' \,|\, z' \neq w'\}. \tag{B.20}$$

Let us then consider the complements

$$U^c = \{(\mathbf{p}, \mathbf{q}, \mathbf{r}) \subset N \,|\, z = w\}, \tag{B.21}$$

$$U'^c = \{(\mathbf{p}', \mathbf{q}', \mathbf{r}') \subset N' \,|\, z' = w'\}. \tag{B.22}$$

We are interested in the projections $p_{13}$ of these sets. We have

$$\dim_\mathbb{R}(p_{13}(U'^c)) \leq 2(n-1) + \max\left(\dim_\mathbb{R}(\mathcal{C}_\Sigma^a) + \dim_\mathbb{R}(\mathcal{D}_\Sigma^b) - 2, 0\right), \tag{B.23}$$

$$\dim_\mathbb{R}(p_{13}(N')) = 2(n-1) + \dim_\mathbb{R}(\mathcal{C}_\Sigma^a) + \dim_\mathbb{R}(\mathcal{D}_\Sigma^b), \tag{B.24}$$

where the second term on the RHS of the first equation is the expected dimension of the intersection $\mathcal{C}_\Sigma^a \cap \mathcal{D}_\Sigma^b$. This dimensional estimate clearly shows that $U^c$ does not contribute to the class of the second term on the LHS of (B.8). On the other hand,

$$\dim_\mathbb{R}(p_{13}(U^c)) \leq 2n, \tag{B.25}$$

$$\dim_\mathbb{R}(p_{13}(N)) = 2(n-1) + \dim_\mathbb{R}(\mathcal{C}_\Sigma^a) + \dim_\mathbb{R}(\mathcal{D}_\Sigma^b). \tag{B.26}$$

The first inequality can be seen as follows. Provided $\mathcal{C}_\Sigma^a \cap \mathcal{D}_\Sigma^b$ is not empty, the divisors in the first and third factors defining $M$ can coincide but be otherwise free, with the extra unique point in $\Sigma(n+1)$ constrained to lie in $\mathcal{C}_\Sigma^a \cap \mathcal{D}_\Sigma^b$. This constraint becomes irrelevant after projecting via $p_{13}$, and the dimension of the diagonal is indeed $2n$. It is also easily seen that, similarly to the previous case, any other configuration must have smaller dimension.

Thus, we can conclude that whenever

$$\dim_\mathbb{R}(\mathcal{C}_\Sigma^a) + \dim_\mathbb{R}(\mathcal{D}_\Sigma^b) = 2, \tag{B.27}$$

and the intersection is not empty, there is a contribution to the class of $\mathcal{E}_{n+1}^a \cdot \mathcal{F}_n^b$ coming from the diagonal that is not compensated by anything in $\mathcal{F}_{n-1}^b \cdot \mathcal{E}_n^a$. Since this can happen only if $a = b$ (given the ordering of the basis $\mathcal{C}_\Sigma^a$ and $\mathcal{D}_\Sigma^b$), we obtain (B.8) up to relative signs. The signs can be fixed by taking into account how the above isomorphism (B.16) changes orientations. The first two relations (B.6) (B.7) are similar but simpler, so we omit the details.

# C  Background flat connections and conformal blocks

The free hypermultiplet can be coupled to a flat connection for the background $SU(2)$ flavour symmetry. The discussion in 3.2.1 carries through with minor changes. All we have to do is to replace $H^{\bullet,\bullet}(\Sigma)$ with

$$H^{\bullet,\bullet}(\Sigma, E), \tag{C.1}$$

for some vector bundle $E$. This changes the number of Heisenberg algebras that arise in our construction. For example, for generic $E$, we get

$$H^{0,0}(\Sigma, E) = 0, \tag{C.2}$$
$$H^{0,1}(\Sigma, E) = g - 1. \tag{C.3}$$

The total dimension of the Hilbert space is then $2^{2g-2}$. The fact that this number agrees with the dimension of the conformal blocks of an algebra of fermionic currents is no coincidence. In fact, the free hypermultiplet enjoys a $(0, 4)$ boundary condition where only the righ-moving fermions are allowed to fluctuate at the boundary. These fermions are precisely the one-form fermions $\chi_\mu$ whose fluctuations span $H^{0,1}(\Sigma, E)$.

# D  Review of the mathematical definition of the Coulomb branch

The basic idea of the mathematical definition of the Coulomb branch is to focus on an infinitesimal neighbourhood of a monopole operator and to mimic, algebraically, its action in the presence of matter fields. Thus, let $D := \mathbb{C}[[z]]$ be the formal disk, $D^* := \mathbb{C}((z))$ be the formal punctured disk, with the idea that a monopole operator is inserted at the origin of $D$. We would like to consider two configurations of gauge and matter fields on $D$ that differ by the insertion of a monopole operator at the origin.

For SQED[1], the construction works as follows. Let $\mathcal{P}$ be a $\mathbb{C}^*$ (the complexification of $U(1)$) algebraic principal bundle over $D$, $t$ a trivialisation of the bundle over $D^*$. Furthermore, let $s$ be a section of $\mathcal{P} \times_{\mathbb{C}^*} N$, where $N \cong \mathbb{C}$ is the fundamental representation, and define the set of triples $\mathcal{T} = (\mathcal{P}, t, s)$. Consider then the pull-back

$$\mathcal{T} \times_{\mathbb{C}((z))} \mathcal{T} = \{(\mathcal{P}_1, t_1, s_1) \times (\mathcal{P}_2, t_2, s_2) \in \mathcal{T} \times \mathcal{T} \mid t_1(s_1) = t_2(s_2)\} / \text{iso}. \tag{D.1}$$

Naïvely, this pull-back parametrises configurations of gauge and matter fields on two disks that differ at the origin. If we require in addition $P_2$ to be trivial and $t_2$ to extend to the origin, then we get

$$\mathcal{R} := \{(\mathcal{P}, t, s) \mid t(s) \in N[[z]]\} / \text{iso}. \tag{D.2}$$

This is the space of triples $(\mathcal{P}, t, s)$ such that the trivialisation of the section $s$ extends to the origin of $D$. It turns out that this space is quite sufficient to describe Coulomb branch operators, and in fact, the Coulomb branch chiral ring is defined as the $\mathbb{C}^*[[z]]$ equivariant Borel-Moore homology of $\mathcal{R}$,

$$\mathbb{C}[M_C] := H_*^{\mathbb{C}^*[[z]]}(\mathcal{R}), \tag{D.3}$$

with a product given by a convolution product

$$H_*^{\mathbb{C}^*[[z]]}(\mathcal{R}) \times H_*^{\mathbb{C}^*[[z]]}(\mathcal{R}) \to H_*^{\mathbb{C}^*[[z]]}(\mathcal{R}), \tag{D.4}$$

whose general definition we omit, but which we are going to concretely describe presently. First, notice that

$$\mathcal{R} = \bigsqcup_{n \in \mathbb{Z}} z^n \mathbb{C}[z] \cap \mathbb{C}[z] \tag{D.5}$$

$$\cong \bigsqcup_{n \in \mathbb{Z}} z^{\max(0,n)} \mathbb{C}[z]. \tag{D.6}$$

As a vector space we can write the equivariant Borel-Moore homology as

$$H_*^{\mathbb{C}^*[[z]]}(\mathcal{R}) \cong \bigoplus_{n \in \mathbb{Z}} H_*^{\mathbb{C}^*}(\text{pt}) \tag{D.7}$$

$$\cong \bigoplus_{n \in \mathbb{Z}} \mathbb{C}[w]. \tag{D.8}$$

We now have to determine the product. Let us denote $x$ and $y$ are the fundamental classes for $n = 1$ and $n = -1$. The product of these classes corresponds to the push-forward

$$z\mathbb{C}[z] \to \mathbb{C}[z], \tag{D.9}$$

which is the cup product of $w$ with the fundamental class. Thus the product of $xy$ is $w$ and so

$$H_*^{\mathbb{C}^*[[z]]}(\mathcal{R}) \cong \mathbb{C}[x, y]. \tag{D.10}$$

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
