# Peer review of "Supersymmetric ground states of 3d $\mathcal{N}=4$ SUSY gauge theories and Heisenberg Algebras"

_SciPost Physics, doi:SciPost Phys. 14, 063 (2023)_

## Round 1 · Referee Report · Anonymous (Referee 1) · 2022-8-29

Report

This is an interesting paper on the supersymmetric quantum mechanics that are obtained by twisted compactifications of three-dimensional N=4 field theories on Riemann surfaces. Depending on the type of topological twist, the operators of the quantum mechanics (QM henceforth) are local Higgs or Coulomb branch operators, together with their topological descendants along the Riemann surface. Important past work on the subject includes the calculation of topologically twisted indices and the study (by the author and collaborators) of the space of ground states of the theory on a Riemann surface.

The main results of this preprint are: - a general derivation of the commutator of the QM operators in terms of the secondary product of the topologically twisted theory and the intersection pairing of the Riemann surface; - an explicit construction of the action of such operators on the Hilbert space of the QM for the two simplest 3d N=4 theories: the free hypermultiplet and its mirror theory SQED[1]. The operators realize Heisenberg algebras. The calculation is fairly straightforward for the free hypermultiplet, and more sophisticated for the mirror SQED[1], in which the operators turn out to be Nakajima's operators. In the latter case the derivation is less direct, but the author provides convincing evidence for the claim by checking that the expected Heisenberg algebra is satisfied.

The paper is generally clearly written, albeit technical and reliant on mathematical jargon in places. The results are interesting, especially the connection to Nakajima's operators, and pave the way for further work in a number of directions, some of which are summarized in the conclusion. This in my opinion justifies the publication of the paper, after a minor revision to correct typos and add a few more explanations for the sake of the non-expert readers. I lay out the requested changes below.

Requested changes

1 - There are a few typos in the text, including one in the abstract. Please run a spell-check. 2 - p. 3: define the hermitian conjugates of the eight supercharges. 3 - p. 4: correct typos: a missing dot in eq. (2.15) and a spurious 'mirror' before Rozansky-Witten above (2.17). 4 - p. 5: upper indices in (2.26); define the 2-sphere $S^2_{x_2}$ in (2.29). 5 - p. 5: a concrete example of the secondary product (perhaps for the free hyper?) would be useful. 6 - p. 7, beginning of the paragraph above (2.42): I do not see how the cycle in (2.41) vanishes if $\mathcal{C}^{1,2}_\Sigma$ are 1-cycles. Please clarify. 7 - p. 8: is there a spurious m in mRW in eq. (3.14)? Same in p. 10. 8 - p. 10, eq. (3.21): explain which field contributes each of the summands, and check the $\mathbb{C}[-f]$ gradings - shouldn't they be opposite to be consistent with Table 1?
9 - p. 11: typos $f_g$ in (3.28) and (3.31). 10 - p. 11, second half: check that the commutator (3.24) is reproduced, sign included. 11 - p. 13, end of paragraph after (4. 16): please work out or justify explicitly the correct constants of proportionality. 12 - p. 14: explain the notions of perverse sheaf and hypercohomology for the sake of non-experts. 13 - p. 15, eq. (4.29): explain which field contributes each of the summands. 14 - p. 16, eqs. (4.34), (4.35): are $\mathbf{p}$ and $\mathbf{q}$ the other way around? 15 - p. 16, after (4.42): include an explicit formula for the descendant(s) of a monopole operator in terms of integrals of monopole operators and gauginos. 16 - p. 17: explain what the asterisks in (4.44) stand for, and expand the last sentence of section 4 to be more explicit and self-contained.

---

## Round 1 · Referee Report · Anonymous (Referee 2) · 2022-9-11

Strengths

  1. Relatively self-contained.
  2. Interesting computation.

Weaknesses

  1. Could be more self-contained. (Some of the mathematics will not be familiar to the typical hep-th reader.)
  2. Deals with a single example.

Report

This is an interesting paper that explore in detail the most basic example of a 3d N=4 mirror symmetry, that between a free hypermultiplet and a U(1) gauge theory with a single charged hyper. It does so in the context of quantising the theory on a Riemann surface, by considering the B/A -twist and constructing explicitly the corresponding N=4 quantum mechanics.

This will be of general interest to experts in supersymmetric QFT, especially for those interested in the interface between susy QFT and mathematical physics.

A few very minor corrections or proposed clarifications are listed below.

I fully recommend this note for publication in SciPost.

Requested changes

  1. after (3.25), perhaps mention this is an anticommutator.

  2. above (4.2), it should be "vector multiplet" instead of "hypermultiplet".

  3. In section 4.1, could the author explain what is the mirror of the twisted fermions \chi_\mu that appeared in the B-model?

---

## Round 2 · Referee Report · Anonymous (Referee 1) · 2022-11-22

Report

The revised version addresses the minor issues raised in my previous report and makes the preprint more readable and self-contained. I recommend the paper for publication in SciPost.

---

## Round 2 · Author Response

I thank the referees for their helpful comments. I addressed them as outlined in the list of changes.

---

## Round 2 · List of Changes

All the points I do not mention below have been addressed without requiring further explanation (please note that I removed an unnecessary equation after 4.13 and so after then the numbering is off by one with respect to the referee's comments).

Report 2:

  1. I introduce the mirror of the twisted fermions in section 4.2. I added a comment in section 4.1 mentioning this, and then emphasised this fact below equation 4.27 (see also 15 below).

Report 1:

  1. This comment made me realise that I was using a non-standard convention for A- and B- twist indices. I consistently changed dotted and undotted indices in 2.1 and 2.2.
  2. I give concrete examples of secondary products in later sections (3.2.2 and 4.1).
  3. The extra sign that caused the confusion was spurious, I thank the referee for pointing this out.
  4. I actually found a similar computation in one of the references I was using, and therefore I cited it. Normalisations can be introduced at various stages (normalisation of the fields, of the brackets, etc.). I work with one that is fixed by mirror symmetry and that ensures that \varphi is the complex moment map for the topological symmetry. This ensures in particular that the monoopole operators have integer charges under this symmetry.
  5. I agree that it is unhelpful to introduce the notion of perverse sheaf, which would take too long to explain in this context. It is better to talk about geometric quantisation, and I do so instead. The geometric quantisation used here reduces to the computation of standard de Rham cohomology of the target. I refer to my previous paper (cited) for further details.
  6. I introduced a schematic expression in 4.38.

Finally, I have corrected a few additional typos, especially in 2.32, 2.34 and the appendices. A wrong equation was referenced after 2.45 and I corrected the discussion accordingly.

---

## Editorial Decision

published